# Inflammatory Bowel Diseases and Non-Alcoholic Fatty Liver Disease: Piecing a Complex Puzzle Together

**DOI:** 10.3390/ijms25063278

**Published:** 2024-03-14

**Authors:** Rossella Maresca, Irene Mignini, Simone Varca, Valentin Calvez, Fabrizio Termite, Giorgio Esposto, Lucrezia Laterza, Franco Scaldaferri, Maria Elena Ainora, Antonio Gasbarrini, Maria Assunta Zocco

**Affiliations:** CEMAD Digestive Disease Center, Fondazione Policlinico Universitario “A. Gemelli” IRCCS, Catholic University of Rome, 00168 Rome, Italy; rossella.maresca12@gmail.com (R.M.); irene.mignini@guest.policlinicogemelli.it (I.M.); simonecv95@gmail.com (S.V.); valentino.calvez@gmail.com (V.C.); fabrizio.termite@libero.it (F.T.); giorgio.esposto2@gmail.com (G.E.); lucrezia.laterza@policlinicogemelli.it (L.L.); franco.scaldaferri@policlinicogemelli.it (F.S.); mariaelena.ainora@policlinicogemelli.it (M.E.A.); antonio.gasbarrini@unicatt.it (A.G.)

**Keywords:** inflammatory bowel diseases, Crohn’s disease, ulcerative colitis, liver steatosis

## Abstract

Inflammatory bowel diseases (IBD), comprising Crohn’s disease and ulcerative colitis, are systemic and multifaceted disorders which affect other organs in addition to the gastrointestinal tract in up to 50% of cases. Extraintestinal manifestations may present before or after IBD diagnosis and negatively impact the intestinal disease course and patients’ quality of life, often requiring additional diagnostic evaluations or specific treatments. Non-alcoholic fatty liver disease (NAFLD) is the most common liver disease worldwide. Current evidence shows an increased prevalence of NAFLD (and its more advanced stages, such as liver fibrosis and steatohepatitis) in IBD patients compared to the general population. Many different IBD-specific etiopathogenetic mechanisms have been hypothesized, including chronic inflammation, malabsorption, previous surgical interventions, changes in fecal microbiota, and drugs. However, the pathophysiological link between these two diseases is still poorly understood. In this review, we aim to provide a comprehensive overview of the potential mechanisms which have been investigated so far and highlight open issues still to be addressed for future studies.

## 1. Introduction

Inflammatory bowel diseases (IBD), comprising Crohn’s disease (CD) and ulcerative colitis (UC), are multifaceted disorders characterized by the recurrent inflammation of the gastrointestinal (GI) tract. The systemic nature of IBD is increasingly recognized, with manifestations often extending beyond the intestinal confines [1]. Extraintestinal manifestations (EIMs) occur in up to 50% of IBD patients and may present before or after the diagnosis of the bowel disease, with a heterogeneous impact on patients’ quality of life, ranging from mild forms to potentially lethal complications [2]. The most commonly affected organs are skin, joints, eyes, and the hepato-biliary system [2]. The natural history of EIMS often follows the activity of the bowel disease and may benefit from therapies targeting intestinal inflammation, with some specific exceptions, notably pyoderma gangrenosum, ankylosing spondylitis, primary sclerosing cholangitis, and uveitis, which usually evolve independently from the IBD course and require multidisciplinary approaches [3]. Although the exact mechanisms underlying EIMs are still unclear, the frequent association of different EIMs supports the hypothesis of a common pathogenesis deriving from the extension of inflammatory processes beyond the GI tract. Indeed, in a recent workshop paper published in 2019, an expert panel from the European Crohn’s and Colitis Organization (ECCO) provides a standardized definition of EIMs as inflammatory disorders in IBD patients “*located outside the gut and for which the pathogenesis is either dependent on extension/translocation of immune responses from the intestine, or is an independent inflammatory event perpetuated by IBD or that shares a common environmental or genetic predisposition with IBD*” [4].

Hepatic involvement has emerged as a prominent EIM, shedding light on the intricate crosstalk between the gut and liver [5]. Among IBD-related liver diseases, primary sclerosing cholangitis is the most fearsome, but other disorders are more common, such as cholelithiasis and non-alcoholic fatty liver disease (NAFLD) [2,5]. NAFLD is defined by the buildup of fat in the liver cells, which cannot be linked to an excessive alcohol intake. It is thought to be a continuum of conditions, from simple hepatic steatosis, where fat accumulation is the primary sign, to non-alcoholic steatohepatitis (NASH), a more severe stage linked to inflammation and liver cell damage [6]. Hepatocellular damage, fibrosis. and inflammatory alterations are characteristics of the transition from basic steatosis to NASH. NASH can progress to cirrhosis, liver failure, or hepatocellular carcinoma (HCC). Because of its high frequency, correlation with metabolic disorders, and potential for severe liver-related consequences, this condition represents a substantial public health burden [7] and should, therefore, not be underestimated as an EIMs in IBD patients. The clinical course of gut inflammation itself seems to be negatively influenced by NAFLD occurrence. In a recent systematic review analyzing the impact of NAFLD on IBD-related hospitalization outcomes, Boustany et al. highlighted how NAFLD may be related to worse outcomes in individuals with IBD, despite the low quality of available data [8]. In addition, in a recently published study, Hyun et al. estimated the presence of liver steatosis by calculating the Hepatic Steatosis Index (HSI) and Fibrosis-4 (FIB-4) in a cohort of 3356 IBD patients and found that steatosis was associated with increased risk of clinical relapse both in UC and CD patients [9].

It is worth noting that a consensus statement on a new nomenclature for fatty liver disease has been recently published. Steatotic liver disease (SLD) was chosen as the umbrella term to encompass the different etiologies of steatosis. The name metabolic dysfunction-associated steatotic liver disease (MASLD) was chosen to replace NAFLD. Several reasons led the experts to change the nomenclature, as explained in detail in the paper by Rinella et al. published in June 2023 [10]. These reasons can be summarized as follows: to reduce the stigma associated with the word “fatty” and to include patients with risk factors for NAFLD, such as diabetes, who consume more alcohol than the strict threshold used to define NAFLD. However, in this review, we will use the term NAFLD, as most studies investigating the relationship between IBD and hepatic steatosis were published prior to the consensus paper.

In the context of IBD, alterations in the gut microbiota, increased intestinal permeability and chronic inflammation are just some of the factors contributing to a cascade of events influencing the liver, in addition to the potential role played by chronic therapies to which IBD patients are subjected [11]. Our review delves into the intricate interplay between IBD and NAFLD, exploring the underlying mechanisms of this facet of the gut–liver axis and analyzing the variables that influence the risk of steatosis in IBD patients. First, we provide an overview of NAFLD prevalence and clinical course in the IBD population. We then address its etiopathogenesis and underlying IBD-specific risk factors, with each subsection focusing on a specific underlying mechanism. Thus, we aim to detail every facet of this largely discussed but still poorly understood topic.

## 2. Prevalence of NAFLD in the General Population and IBD Patients

The prevalence of NAFLD in the general population varies worldwide and is influenced by factors such as age, sex, ethnicity, and the presence of metabolic risk factors [12]. According to a meta-analysis published by Younossi et al. in 2022, the overall global prevalence is about 30%, with the highest percentage reported in Latin America (44.37%), while in North America it is about 31% and in Europe about 25% [13]. The prevalence tends to increase with age, and it is higher in individuals with obesity, type 2 diabetes, and metabolic syndrome [14].

Several studies have been conducted analyzing the prevalence of NAFLD in IBD patients, with a fair amount of heterogeneity in the results, probably attributable to the different ways in which steatosis is diagnosed. In addition, the increasing use of biologic drugs of various classes and their possible impact on hepatic steatosis could affect the incidence data.

The gold standard for the diagnosis of hepatic steatosis is still liver biopsy, but this is an invasive procedure that should be reserved only for patients in whom there is a strong clinical suspicion of steatohepatitis or a need for a differential diagnosis. A retrospective cross-sectional study conducted by Bosch et al. used histology as a diagnostic technique for NAFLD, showing a prevalence of 26%, consistent with that of the general population [15]. Another recent study by Dias et al. retrospectively enrolling IBD patients who underwent liver biopsy reported NAFLD as the most frequent etiology of altered liver function tests in IBD (32.7% of patients) [16]. Currently, less invasive methods with high diagnostic accuracy find wide application in clinical practice and have been used for prevalence studies. In a retrospective study conducted in 2020 by Hoffmann et al. on 694 IBD patients, the prevalence of hepatic steatosis assessed by ultrasonography was 46.5% [17]. Such a result represents a striking difference from the general population, belied instead by an American retrospective study from the same year, in which a cohort of 1672 IBD patients reported a prevalence of hepatic steatosis detected by ultrasound of 12.4% [18]. More novel and advanced diagnostic tools, such as magnetic resonance proton density fat fraction and controlled attenuation parameter elastography (CAP-E), showed an increased prevalence of steatosis in IBD patients, both when analyzed by magnetic resonance imaging (MRI) (54.6% in 130 patients) [19] and CAP-E (32.8% in 384 patients) [20].

Considering the prevalence of hepatic steatosis in CD and UC separately, current data appear to be contrasting. Some evidence highlights that CD is a risk factor for the development of hepatic steatosis [18,21,22], and in a large recent meta-analysis, about 67% of IBD patients with NALFD were affected by CD vs 32% who were affected by UC [23]. Conversely, a systematic review conducted by analyzing 13 studies for UC and 4 studies for CD showed a prevalence of up to 55% for patients with UC, and up to 39.5% for patients with CD [5]. Similar data were also reported in the European consensus on EIMs in IBD [2].

Certainly, further studies with less heterogeneity in the population analyzed and with standardized diagnostic methods are needed to clarify the exact prevalence of NAFLD in IBD.

## 3. NAFLD Clinical Course in IBD Patients

The natural history of NAFLD is characterized by the progressive evolution to NASH, liver fibrosis, and eventually cirrhosis, with an increased risk of HCC. It is noteworthy that NAFLD and NASH themselves are risk factors for HCC development, even in the absence of advanced fibrosis or cirrhosis. Whether IBD represents a risk factor for a quicker progression of NALFD to more advanced liver damage compared to the general population is an intriguing matter of study, due to the potential role of the chronic inflammatory status in promoting liver disease. Recently, limited data have begun to appear in the scientific literature.

The prevalence of liver fibrosis in the IBD population varies in different studies, reaching up to 16.7% [24], and there is some evidence revealing higher percentages among NAFLD patients with IBD compared to those without IBD [25,26]. In the aforementioned retrospective American study by Ritaccio et al., liver fibrosis was assessed non-invasively through the NALFD fibrosis score (NFS), which includes age, body mass index, diabetes, platelet count, transaminase levels and serum albumin. Meanwhile, as we previously mentioned, NAFLD prevalence in their cohort of IBD patients was 12.4%, and fibrosis appeared to be much less common. Of a total of 138 patients with sufficient available data to calculate NFS, 4% presented advanced fibrosis at diagnosis and 9% experienced a worsening of fibrosis degree at the 5 year follow-up, suggesting that, despite the high prevalence of NAFLD in IBD patients, progression to more severe liver injury is quite rare [18]. A systematic review recently published by Martínez-Domínguez et al. confirmed a significant, although slight, difference concerning the prevalence of NAFLD in IBD patients (34.1%) vs. the general population (31.2%). Moreover, the authors found NASH to be globally present in about 6% of IBD patients, while cirrhosis prevalence ranged from 0% to 10%. Predictably, concomitant primary sclerosing cholangitis was a major risk factor for cirrhosis occurrence [27]. Unfortunately, data concerning the risk of HCC development in IBD subjects with NAFLD are still lacking [27].

In addition to primary sclerosing cholangitis and to the metabolic features which are well known for being associated with hepatic damage, CD seems to be an independent risk factor for liver fibrosis. Aggarwal et al. examined a population of patients with biopsy-proven NAFLD and concomitant IBD, observing that liver fibrosis and NASH were more common in CD than UC. After multivariate analysis, they concluded that CD is independently associated with histological fibrosis [26].

Surprisingly, although IBD seems to increase the risk of liver disease progression, a Korean study highlighted how only hepatic steatosis but not fibrosis is associated with poor clinical outcomes. More specifically, the authors found that liver steatosis increased the risk of IBD relapse (defined as IBD-related hospitalizations or surgical interventions or the need to start treatments with corticosteroids, immunosuppressors, or biologic drugs) while fibrosis did not, both in CD and UC [9]. Therefore, the link between IBD and NAFLD appears to be even more complex, and the real burden of hepatic steatosis on IBD morbidity needs to be better defined.

## 4. Pathogenetic Mechanisms of NAFLD in IBD

The pathogenetic mechanisms underlying IBD and NAFLD have been the subject of study and have been of great interest to the scientific community for years.

First, UC and CD are notably associated with abnormal and chronic inflammation in the gut. This inflammation derives from a complex network of interactions among genetic, microbial, and environmental factors and the immune system’s response [28]. Regarding microbial factors, the literature has widely described a close relationship between alterations in gut microbiota composition, known as dysbiosis, and IBD [29]. At the same time, the innate and adaptive immune systems are both involved in the pathogenesis of IBD [30]. The innate immune response in the gut, including macrophages and neutrophils, releases pro-inflammatory cytokines like Tumor necrosis factor-alpha (TNFα) and interleukin (IL)-1β, triggering inflammation and recruiting other immune cells to the gut [30]. The adaptive immune system, primarily composed of T and B lymphocytes, is also involved in IBD pathogenesis. For instance, T helper (Th) lymphocytes can differentiate into subtypes such as Th1, Th2, Th17, or regulatory T cells (Treg). Each of these may influence the intestinal production of pro-inflammatory cytokines, such as interferon-gamma (IFN-γ) and IL-17, determining the inflammation cascade at the basis of IBD [31].

Hepatic steatosis is characterized by an imbalance in lipid acquisition and disposal within hepatocytes, resulting from increased fatty acid uptake and impaired fatty acid oxidation or secretion [32]. Furthermore, the dysregulation of lipid metabolism-related transcription factors and signaling pathways contributes to the pathogenesis of hepatic steatosis. For example, sterol regulatory element-binding proteins (SREBPs) are key transcription factors that regulate the expression of lipid synthesis and uptake genes. The activation of SREBPs promotes hepatic lipogenesis and contributes to lipid accumulation in the liver [33]. Additionally, peroxisome proliferator-activated receptors (PPARs) and liver X receptors (LXRs) are nuclear receptors that regulate lipid metabolism and inflammation in the liver. The dysregulation of these nuclear receptors can disrupt lipid homeostasis and exacerbate hepatic steatosis [34].

Despite a large amount of data about NAFLD and IBD, factors linking these two conditions are not yet completely understood. Nevertheless, some pathogenetic mechanisms have been identified and risk factors for NAFLD in IBD patients may be broadly divided into two categories: IBD-related and non-IBD-related factors. Regarding the second group, metabolic diseases including type 2 diabetes, metabolic syndrome, hypertension, obesity, and insulin resistance are well known to promote NAFLD development [35]. In a study analyzing NAFLD in IBD, Glassner et al. found that NAFLD was associated with lower metabolic risk in IBD patients than NAFLD-only patients. In particular, the authors found that NAFLD-only patients had more frequent hypertension, dyslipidemia, diabetes, and higher weight than IBD-NAFLD patients, thus suggesting that factors other than the metabolic profile may contribute to NAFLD occurrence in the IBD population [36]. In a review published in 2021, Papaefthymiou et al. hypothesized that IBD-related risk factors play a greater role than non-specific ones in the pathogenesis of NAFLD in patients with active or severe disease [11]. Inflammation is one of the main IBD-specific factors and drives both the intestinal disease and the hepatic manifestation. Indeed, NAFLD itself, as a facet of metabolic syndrome, can be considered a consequence of chronic systemic inflammation [37].

A further discussion of the metabolic conditions promoting NAFLD is beyond the purpose of our review. Conversely, the complex puzzle of different IBD-specific risk factors is schematically represented in Figure 1 and thoroughly analyzed in the next paragraphs.

### 4.1. Disease Activity

Available data concerning disease activity as an IBD-specific risk factor for NAFLD development are contradictory.

As previously mentioned, Bessisow et al. found that NAFLD was predicted by active disease [38]. In particular, they hypothesized that the underlying link between these two conditions was the increased intestinal permeability, which characterizes the active phase of intestinal disease, and the subsequent translocation of inflammatory products of the intestinal microbiota. Their results were also in accordance with those of Likhitsup et al., who conducted a cross-sectional study and observed that patients with a higher Crohn’s disease activity index (CDAI) had a 1.6-fold increased risk of developing steatosis (odds ratio—OR 1.6 95% *p* = 0.03). They did not find any correlation with UC activity [21]. Contrastingly, Magrì et al. also separated UC and CD in subgroup analysis without achieving any statistically significant difference between NAFLD and non-NAFLD patients in terms of disease activity (in UC, OR 0.62, *p* = 0.20; and in CD, OR 1.70, *p* = 0.31). Particularly, none of IBD-specific risk factors showed an association with NALFD, thus suggesting that the patient’s metabolic profile has a greater influence on steatosis development compared to intestinal inflammation [39]. Similarly, Sagami et al. conducted a retrospective study, including about three hundred CD patients exclusively. They observed, in an interesting way, that NAFLD was even an independent predictor of lower C-reactive protein levels and a higher rate of the remission of disease (OR, 2.57; 95% CI, 1.21–5.80), suggesting almost a hypothetical protective role of NAFLD in CD patients. Specifically, they postulated that the occurrence of NAFLD could indicate an improved general clinical state of the patients—such as a higher body mass index (BMI) or improved nutritional status—and consequently be linked to lower activity of chronic bowel disease [40]. Lastly, in a large prospective case–control study, the authors found no differences in steatosis prevalence between UC remission patients versus mild-activity patients according to the partial Mayo score (17.5% vs. 19.7%, *p* = 0.69) [41].

### 4.2. Surgery

In a retrospective analysis of a cohort of 321 patients with IBD, disease duration and prior surgery for IBD were found to be independent predictors of NAFLD development with an adjusted hazard ratio (HR) of 1.12 (1.03–1.23), *p* = 0.010 and 1.34 (1.04–1.74) *p* = 0.020, respectively, via multivariate cox regression analysis [38].

These findings are confirmed by several studies and by a large meta-analysis of 27 studies and 7640 IBD patients. Lin et al. identified disease duration and bowel resection as IBD-related risk factors with a mean difference of 1.59 years (95% confidence interval—CI: 0.66–2.54, *p* < 0.001) and with an OR of 1.39 (1.01–1.93, *p* < 0.04). The authors discussed possible reasons for the association between a history of bowel surgery and the development of NAFLD, including the direct impact of a short bowel, or dietary changes in patients who are in remission after surgery who might increase their caloric intake and are more likely be obese. It could also be hypothesized that the patients who require surgery are the ones with greater inflammation; systemic inflammation ia a reasonable risk factor for NAFLD, even though its specific role has not yet been fully elucidated [23]. Another finding of the meta-analysis by Lin et al. is that the majority of IBD patients with NAFLD had CD rather than UC (67% vs. 32%). This could be explained by CD patients being more likely to have multiple small bowel inflammations and undergo small bowel surgery, which can potentially alter bile acid metabolism in the ileum. This can lead to decreased ileal farnesoid X receptor (FXR) gene expression and lower levels of circulating fibroblast growth factor 19 (FGF-19). FGF-19 is believed to have downstream effects on lipid metabolism and regulation and bile acid synthesis by the liver [42]. The FXR pathway has been described to be one of the mechanisms involved in the development of NAFLD. Indeed, there are currently clinical trials using FXR agonists, obeticholic acid, for example, as a treatment for NAFLD/NASH [43]. Moreover, recent data also suggest that treatment with FXR agonists reduces inflammation in mouse models of colitis [37].

In vivo preclinical studies on mice have demonstrated that small bowel resection is strongly associated with the development of hepatic steatosis [44]. Barron’s experiments in mice showed that small bowel resection was associated with weight gain and a different body composition, with increased fat mass and decreased lean mass. Fasting blood glucose and TNFα levels were also increased. In addition, small bowel resections result in a significant loss of the endocrine component of the intestine. In response to food ingestion, the gut releases the so-called incretins, glucose-dependent insulinotropic peptide (GIP) and glucagon-like peptide-1 (GLP-1), which are hormones that modulate the insulin secretory response. GIP and GLP-1 are produced and released by enteroendocrine K and L cells located in the small intestine and ascending colon. In addition to having effects on pancreatic beta cells, incretins have extrapancreatic effects, such as promoting glucose entry into cells, regulating fatty acid metabolism, inhibiting gastric emptying, reducing food intake, inhibiting glucagon secretion, and a number of other functions [45]. It is credible that the reduction in enteroendocrine K and L cells that occurs following intestinal resections can alter glucose metabolism and promote metabolic syndrome and the development of fatty liver.

Figure 2 represents intestinal molecular pathways with a protective role against fatty liver, which can be affected by resective surgery, leading to NAFLD development.

### 4.3. Enteral and Parenteral Nutrition

As previously mentioned, IBD patients may have severe nutritional deficits due to either malabsorption or short bowel syndrome (SBS) after surgery. Parenteral nutrition (PN) may be required in acute situations or as chronic treatment and can be a truly life-saving therapy. However, several potential side effects may occur [46,47]. PN nutrition is associated with intestinal and hepatic damage through the atrophy of the gut mucosa and progressive liver damage, often characterized by cholestasis in the early stages [48]. The development of PN adverse effects, such as fatty liver, elevated bilirubin levels, the appearance of biliary sludge, and cholecystitis, can be highly variable, depending on multiple factors including the patient’s age or disease state [49]. SLD and elevated transaminases may occur within fourteen days of starting PN.

The development of NALFD during PN is a result of several mechanisms that can be divided into two categories: direct mechanisms of hepatic damage related to the composition of PN itself and indirect mechanisms resulting from gut mucosal atrophy.

Regarding the direct mechanisms, early PN studies showed how administering high concentrations of dextrose with minimal or no lipid component led to increased lipid synthesis by the liver and, thus, to the development of steatosis [50]. However, even with controlled dextrose preparations, the lipids present can cause SLD. This is because PN lipid emulsions do not follow the same micelles circulation as during enteral nutrition, making the liver the first passage for digestion and likely increasing lipid deposition in the liver [51]. Preclinical studies have convincingly shown how the development of parenteral-associated hepatic steatosis (PNAHS) is linked to events mediated by the suppression of AMP-activated protein kinase (AMPK). Moreover, researchers recently identified a key protein, protein phosphatase 2a (PP2A), which would seem crucial for the pathogenesis of PNAHS [52].

Researchers have recently developed a new term to define liver damage that occurs due to the atrophy of the intestinal mucosa: intestinal failure-associated liver disease (IFALD), often used in place of “parenteral nutrition-associated liver disease” (PNALD). This highlights the importance of the gut and its role in liver damage. IFALD, including SLD, may have a wide spectrum of manifestations, from the simple elevation of liver enzymes to severe outcomes such as the development of fibrosis/cirrhosis [48]. Demonstrating the importance of the gut–liver connection, it has been shown that even small amounts of enteral nutrition (EN) can minimize the negative effects of PN [53]. Pathogenesis involves several mechanisms. As mentioned above, the FXR-FGF19 signaling pathway plays a critical role in modulating hepatic glycemic and lipid control [54]. There is evidence showing that, during EN, FXR-FGF19 signaling regulates hepatic bile acid synthesis and lipid and glucose metabolism [55,56]. In fact, studies using large bowel resection in mice (without EN) have revealed that there is an insufficient activation of gut FXR, resulting in liver damage [51]. Moreover, the use of an enteral FXR agonist has proven to be a hepatic protection [57].

Furthermore, various studies have demonstrated that gut mucosa atrophy leads to increased gut permeability during PN, through mechanisms which are not yet well defined. Data suggest the involvement of a G protein-coupled receptor (TGR5)–glucagon-like peptide 2 (GLP-2) axis [48,58,59,60]. In addition, in PN, there is an alteration in the gut microbiota [61] and associated intestinal inflammation [62], resulting in increased gut permeability [63,64]. As discussed in the previous section, these changes can lead to systemic inflammation, resulting in an increased risk of developing hepatic steatosis. Lastly, increased levels of IL-6 and TNFα have been documented in the course of PN [65,66].

Although, to our knowledge, there are no studies showing an increased incidence of steatosis in IBD patients receiving PN, given the aforementioned mechanisms involved in the development of PNAHS, we can reasonably assume that these also occur in IBD patients receiving PN.

### 4.4. Gut Microbiota

The disruption of the intestinal barrier is recognized as one of the earliest hits in IBD pathogenesis, with a large amount of data from animal models supporting this hypothesis, and gut microbiota alterations in IBD are widely documented [67,68]. Globally, IBD patients show a higher relative abundance of Bacteroidetes and Proteobacteria, while Firmicutes appear reduced compared to healthy subjects [69]. Such a microbiota profile leads to an unbalanced equilibrium between anti-inflammatory and pro-inflammatory bacterial metabolites, with lower levels of protective short chain fatty acids [69] and the increased production of lipopolysaccharide (LPS) [70].

As well as IBD, NAFLD is also associated with increased intestinal permeability and dysbiosis [71] and growing evidence suggests that gut microbiota may play a role in NAFLD occurrence and progression and different microbiota signatures may characterize different NAFLD stages [72,73]. In a very recent Chinese study, Mai et al. retrieved data about 16S ribosomal RNA (rRNA) gene amplicon sequencing from previous studies on NAFLD patients and re-analyzed them through bioinformatic methods. A total of 12 studies and 1189 subjects were included. They found a decrease in α-diversity with NAFLD onset and progression and described specific microbiota profiles associated with liver steatosis, fibrosis, and cancer, thus proposing to use some crucial genera, such as *Desulfovibrio*, *Negativibacillus*, and *Prevotella*, as predictive biomarkers for NAFLD [74]. Interestingly, when analyzing microbiota functions, they detected higher levels of LPS and enhanced tryptophan, glutathione, and lipid metabolism in NAFLD patients [74].

Therefore, gut microbiota and its products have been hypothesized to represent a possible link between IBD and NAFLD. LPS, for instance, is a well-known pro-inflammatory factor that is increased in IBD patients and proved to promote insulin resistance and diabetes, further confirming the existence of an inflammatory trigger underlying metabolic disorders and supporting a causative relationship between intestinal dysbiosis and NAFLD [75]. Furthermore, in a recent study on mice with dextran sodium sulfate (DSS)-induced colitis, Kwon et al. observed an impaired intestinal barrier function with increased pro-inflammatory cytokines, elevated LPS circulating levels and a more significant hepatic fat accumulation. The link between these two conditions was found in alterations in lipid metabolism, mainly through the dysregulation of crucial metabolism controllers, including sirtuin 1, adiponectin, fibroblast growth factor 21, and irisin [76]. Globally, these results suggest that the translocation of bacterial products and inflammatory mediators from the bowel to the liver through the portal circulation may predispose a person to NAFLD development.

### 4.5. IBD Medications

The therapeutic armamentarium for IBD is rapidly expanding and, in addition to conventional anti-inflammatory treatments such as aminosalicylates, corticosteroids, and immunosuppressors, it currently also encompasses an increasing number of biological drugs and small molecules. The pathogenetic mechanisms through which different therapies may positively or negatively influence NAFLD pathogenesis are complex and multifaceted. While drugs may exert a potential protective effect against NAFLD by directly reducing systemic inflammation and intestinal disease activity, on the other hand, IBD remission itself could predispose a person to weight gain and, thus, to metabolic syndrome and insulin resistance, a consequence which has been especially described in CD patients [11,77].

Therefore, to date, available data about the role of medications in the development of NAFLD are still contradictory. According to a recent meta-analysis by Lapumnuaypol et al., there is no significant association between IBD drugs and the possibility of developing NAFLD [78]. They included seven observational studies with a whole of about 1600 patients. More specifically, the authors found no publication bias and obtained OR for NAFLD in patients who underwent biological agents or steroids of 0.85 and 1.24, respectively. Similarly, in an Italian study including 465 IBD patients and 189 subjects with gastrointestinal non-IBD disorders, Principi et al. reported a higher prevalence of NAFLD in the IBD group, but no significant association between the hepatic disease and IBD drugs. However, they noted that IBD patients with NAFLD were younger than non-IBD ones (49.9 ± 15.0 vs. 56.2 ± 12.1 years, *p* value = 0.02). They hypothesized that such a result may be a consequence of an early and long-term intake of drugs or the frequent self-medication habit in IBD patients. Thus, the younger age of NAFLD onset may be an indirect sign of the negative impact of IBD drugs on liver disease [41].

Corticosteroids are a cornerstone in IBD treatment and they are recommended by international guidelines to induce disease remission in moderate-to-severe cases, both in CD and in UC [79,80]. Patients with frequent disease relapses often experience prolonged and high-dose steroid use, which entails an increased risk of endocrinological and metabolic disorders [81]. However, several studies have demonstrated no association between steroid use and NAFLD. Thus, Magrì et al. conducted an observational study to evaluate the presence of liver steatosis in IBD and its association with disease-specific risk factors. They observed no association with the use of steroids (OR = 0.49, *p* value = 0.57) nor biologic therapy (OR = 0.78, *p* = 0.41), confirming that the risk factors in these patients were comparable to those found in the general population [39]. Similarly, in a recent systematic review and meta-analysis, including 12 observational studies, Trivedi et al. found no association between steroid use and NAFLD development in IBD patients [82]. Conversely, in a study by Sourianarayanane et al., the use of steroids (prednisone and budesonide) proved to be an independent factor associated with NAFLD (OR = 3.7) both through univariate and multivariate analysis [83]. Interestingly, in contrast with the aforementioned results by Principi et al. [41], the authors observed that IBD patients who presented with NAFLD were older than controls [83]. However, due to the lack of agreement in the literature, more research is required to better understand the potential involvement of steroids in NAFLD development in IBD patients.

To date, no study has shown a direct correlation between NAFLD and other immunomodulatory drugs, particularly methotrexate (MTX), in IBD population [84]. MTX is commonly used for the induction or maintenance of disease remission and is often associated with alterations in liver enzymes (up to 15–50% of patients) [85]. However, rather than causing NAFLD, it exerts its hepatotoxic effect mainly by inducing liver fibrosis (with an estimated risk of 5%), especially in the context of long-term and high-dose use [86], even if this risk seems also to be overestimated according to recent evidence [87]. Indeed, a multicenter longitudinal British cohort study investigated the impact of long-term MTX therapy on liver fibrosis. About one thousand rheumatologic patients were included and liver fibrosis was assessed by non-invasive tests (transient elastography) and biomarker measurements (enhanced liver fibrosis). Diabetes proved to be the most statistically significant risk factor associated with fibrosis, while there was no correlation between the duration of therapy and the cumulative MTX dose and liver fibrosis [87]. To date, there is scarce evidence regarding MTX treatment as a risk factor for the development of NAFLD in IBD patients. To the best of our knowledge, only a recent meta-analysis by Zou et al. confirmed MTX as an IBD-specific risk factor for NAFLD with an OR of 1.76 (CI 95%, 1.02–3.06) [35].

In the last decade, the introduction of biological therapy in IBD has radically changed the course of the disease. From the first biologic agent (infliximab) to the current small molecules, several drugs with different mechanisms of action have been developed over the years, aiming at increasingly personalized medicine. TNFα, one of the main inflammatory cytokines involved in IBD pathogenesis, seems to be associated with NAFLD and its severity [88]. Notably, in a recent meta-analysis by Potoupni et al., the authors observed higher levels of TNFα in patients with histologically confirmed NAFLD compared to healthy controls [88]. TNFα also inhibits adiponectin, an adipokine with anti-inflammatory and anti-steatosis effects [89]. Indeed, it seems that in IBD patients, lower levels of adiponectin correlate with a more aggressive disease phenotype [90]. Similarly, it is known that mesenteric visceral fat is altered both morphologically and functionally in patients with IBD, especially in patients with CD [91]. Specifically, mesenteric and creeping fat CD patients seem to produce higher levels of pro-inflammatory cytokines and adipokines [91].

For all the aforementioned reasons, several studies have investigated the potential role of biological anti-TNFα drugs on NAFLD with contrasting results. In a retrospective study, Bessisow et al. observed that the development of NAFLD was not associated with anti-TNFα therapy (HR 1.69, *p* = 0.056). Interestingly, NAFLD was predicted by disease activity, with an adjusted HR of 1.58 (CI 1.08–2.33, *p* = 0.02) [38]. Conversely, in a cross-sectional study by Likhitsup et al. including 80 IBD patients under treatment with TNFα agents, the authors, assuming a general prevalence of 30%, observed a significantly higher prevalence of NAFLD in these patients (*p* < 0.001), but they did not find any associations between disease duration and NAFLD [21]. Also, another study showed a greater incidence of NAFLD in patients not receiving anti-TNFα therapy (*p* = 0.048), suggesting even a possible protective role of these medications [83].

Regarding the other biological drugs, clinical data are scarce. For instance, there are no studies that have directly and primarily investigated the role of vedolizumab (anti-α4β7 integrin) in NAFLD. In a Taiwanese retrospective study, NAFLD-IBD patients were more frequently treated with vedolizumab than those without NAFLD (16.7% vs. 1.8% *p* = 0.025), but investigating vedolizumab treatment was not the primary aim of the study [92]. Similarly, minimal data about anti-IL 12/23 are available, mostly deriving from non-IBD studies. In particular, treatment with anti-IL12/23 can up-regulate leptin, an adipokine with a positive effect on visceral adiposity. Leptin has an anti-steatosis impact in the early stages of NAFLD and, instead, may later promote the disease’s progression through its fibrogenic action [93]. Lastly, to the best of our knowledge, there are no clinically relevant data on Janus Kinase (jak)-inhibitors or other small molecules.

The pathogenetic mechanisms underlying NAFLD onset in IBD patients, which are widely discussed in the previous sections, are summarized in Table 1.

## 5. Conclusions and Future Directions

In conclusion, SLD is a frequent EIM of IBD, potentially associated with an increased risk of intestinal disease relapse and worse clinical outcomes. Despite a great number of studies investigating the association between these two conditions, data about the real burden of liver steatosis in IBD and the specific molecular mechanisms linking liver and gut inflammation are confusing. Evidence concerning the prevalence and clinical course of liver steatosis is strongly encumbered by the lack of standardized diagnostic tools, which makes it hard to reliably compare results from different studies. Pathogenesis is multifactorial, including both general and IBD-related risk factors. It is recognized that pharmacological and surgical treatments, gut microbiota, parenteral nutrition, disease activity and duration may play a role in SLD occurrence in IBD patients, but specific molecular pathways are not completely understood, and many questions remain unanswered. Finally, the new definition of MASLD, which replaces the previous NAFLD, makes such a context still more complex and an update of current available data according to the novel recommendations is urgently needed.

## Figures and Tables

**Figure 1 ijms-25-03278-f001:**
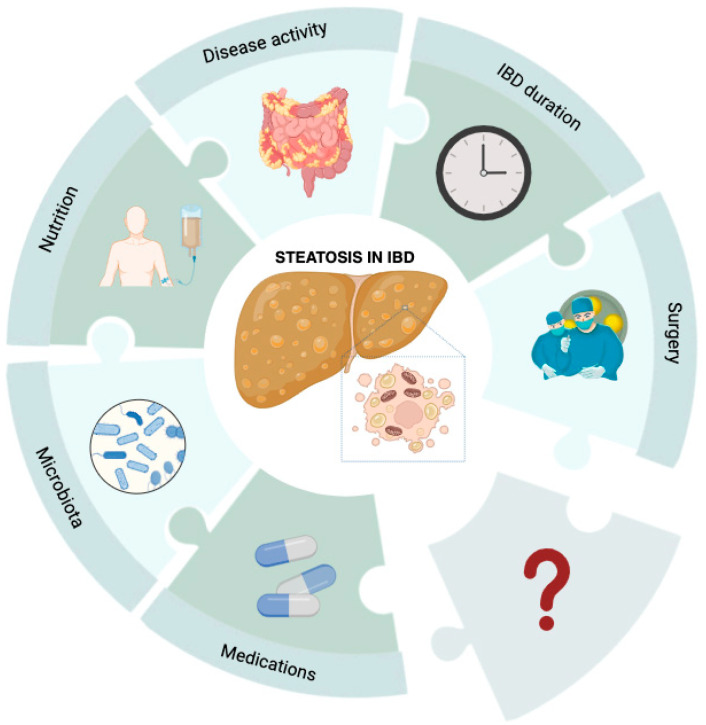
IBD-specific risk factors for liver steatosis. Steatotic liver disease in IBD patients results from a series of factors that are not yet fully elucidated. Drugs, gut microbiota, parenteral nutrition, disease activity, and duration, as well as surgical resections, may yield an effect on liver disease, but many questions about liver steatosis pathogenesis in this subset of patients are still unanswered.

**Figure 2 ijms-25-03278-f002:**
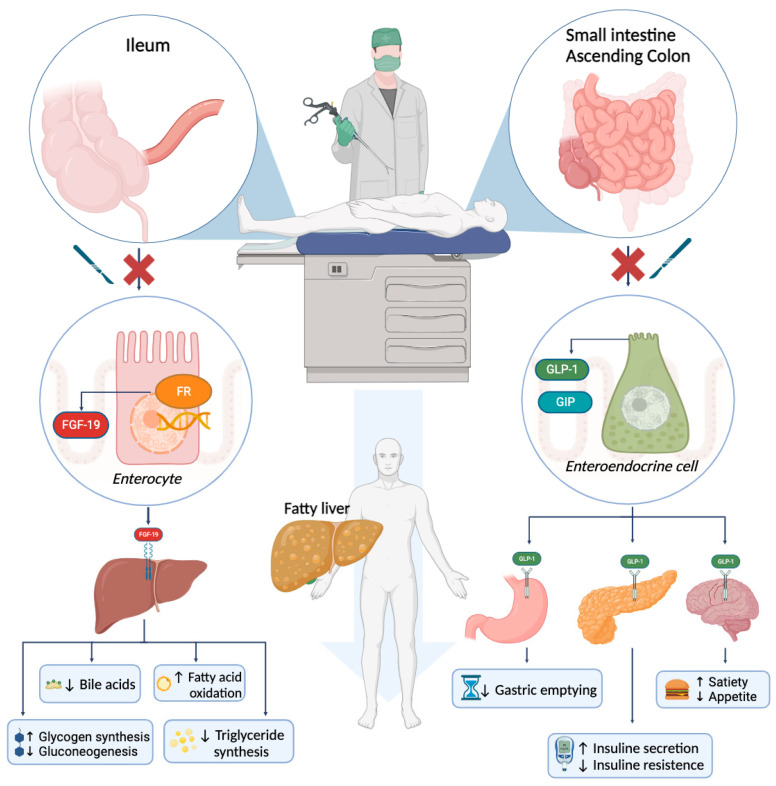
Intestinal surgery impact on liver steatosis. Enterocytes and enteroendocrine cells are deeply involved in regulating bile acids, glucose, and lipid metabolism. Surgical resection of intestinal tracts proved to alter such metabolic functions, finally ending in liver steatosis (central arrow). Further molecular details are provided in the main text.

**Table 1 ijms-25-03278-t001:** Pathogenetic mechanisms of NAFLD in IBD patients. As discussed throughout the text, different IBD-specific risk factors act through multifaceted mechanisms in promoting NAFLD onset. Unfortunately, some specific molecular pathways still need to be clarified.

Risk Factors	Pathogenetic Mechanisms
Disease Activity	Increased intestinal permeability and translocation of inflammatory products Systemic inflammation with increased level of circulating proinflamatory cytokines (e.g., TNF-alpha, IL-6)
Surgical Resections	Resection of the ileum results in a reduction in circulating FGF19 levels, subsequently leading to alterations in bile acid metabolism, glucose metabolism, reduced fatty acid oxidation, and increased triglyceride synthesis Resection of the small intestine and ascending colon leads to reduced circulating levels of GLP-1 and GIP, resulting in accelerated gastric emptying, heightened insulin resistance, and increased appetite
Enteral and Parenteral Nutrition	PN can cause hepatic damage through direct mechanisms (e.g., lipid deposition due to PN lipid emulsions) and indirect mechanisms (e.g., gut mucosal atrophy) EN may mitigate the negative effects of PN on liver health by activating protective molecular pathways (e.g., FXR–FGF19 pathway, TGR5–GLP2 pathway)
Gut Microbiota	Dysbiotic unbalanced equilibrium between anti-inflammatory and pro-inflammatory bacterial metabolites lower the levels of protective short-chain fatty acids (SCFAs) and increase production of lipopolysaccharide (LPS) Increased intestinal permeability and translocation of inflammatory products
IBD Medications	Anti-TNF-alpha treatments may reduce systemic inflammation and intestinal disease activity but may also predispose to metabolic syndrome and insulin resistance, especially in CD patients Data on the association between specific medications (e.g., corticosteroids, biological agents) and NAFLD are conflicting and require further investigation

## Data Availability

No new data were created or analyzed in this study. Data sharing is not applicable to this article.

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
