# Peer review of "Inflammatory Bowel Diseases and Non-Alcoholic Fatty Liver Disease: Piecing a Complex Puzzle Together"

_ijms, 2024, doi:10.3390/ijms25063278_

Round 1

Reviewer 1 Report

Comments and Suggestions for Authors

The review provides a comprehensive exploration of the intricate relationship between Inflammatory Bowel Disease (IBD) and Non-Alcoholic Fatty Liver Disease (NAFLD). Acknowledging the systemic nature of IBD, encompassing Crohn's disease and ulcerative colitis, the paper highlights the significant impact of extra-intestinal manifestations on patients' quality of life. It sets the stage by emphasizing the increased prevalence of NAFLD, particularly its advanced stages, in individuals with IBD compared to the general population.

The review delves into various hypotheses regarding the etiopathogenetic mechanisms linking IBD and NAFLD. It meticulously discusses potential factors such as chronic inflammation, malabsorption, previous surgical interventions, alterations in fecal microbiota, and pharmacological treatments, and sheds light on the complex interplay between these two conditions. 

Overall, the review is well-structured and informative, offering a comprehensive overview of the topic. However, a few suggestions for improvement include:

1. Addition of a table/illustration summarizing the pathogenetic mechanisms as a visual aid for the reader

2. Some discussion on molecular mechanisms involved in the two conditions would also be beneficial 

Author Response

The review provides a comprehensive exploration of the intricate relationship between Inflammatory Bowel Disease (IBD) and Non-Alcoholic Fatty Liver Disease (NAFLD). Acknowledging the systemic nature of IBD, encompassing Crohn's disease and ulcerative colitis, the paper highlights the significant impact of extra-intestinal manifestations on patients' quality of life. It sets the stage by emphasizing the increased prevalence of NAFLD, particularly its advanced stages, in individuals with IBD compared to the general population. The review delves into various hypotheses regarding the etiopathogenetic mechanisms linking IBD and NAFLD. It meticulously discusses potential factors such as chronic inflammation, malabsorption, previous surgical interventions, alterations in fecal microbiota, and pharmacological treatments, and sheds light on the complex interplay between these two conditions.

Overall, the review is well-structured and informative, offering a comprehensive overview of the topic. However, a few suggestions for improvement include:

  1. Addition of a table/illustration summarizing the pathogenetic mechanisms as a visual aid for the reader

R: we thank the reviewer for the suggestion: we have added a table summarizing the pathogenetic mechanisms discussed throughout the text.

  1. Some discussion on molecular mechanisms involved in the two conditions would also be beneficial

R: according to the reviewer’s comment, we have extended the section 4 including an introduction about molecular mechanisms underlying IBD and NAFLD.

Reviewer 2 Report

Comments and Suggestions for Authors

This is an interesting and well written review article that focuses on providing an in-depth overview of the potential mechanisms that have been investigated but also highlight issues that need to be addressed for future studies in linking the pathophysiology of IBD and NAFLD. The authors have discussed in detail the prevalence of NAFLD in the general population and IBD patients, the clinical course of NAFLD in IBD patients and the mechanisms underlying the pathophysiology of NAFLD in IBD. However, the authors agreed that challenges still remain in understanding the link between NAFLD and IBD and despite a great number of studies investigating the association between these two disease conditions, data about the real burden of liver steatosis in IBD and the specific molecular mechanisms linking liver and gut inflammation are still confusing and less understood. The authors have further stated that due to the lack of standardized diagnostic tools, it becomes very difficult to rely on results from different studies that are based on evidence concerning prevalence and clinical course of the liver steatosis. The authors finally concluded that although various factors including pharmacological and surgical treatments, gut microbiota, parenteral nutrition, disease activity and duration could play a role on SLD (steatosis liver disease) occurrence in IBD patients, specific molecular pathways involved are not completely understood and many questions still remain unanswered. Moreover, readers will also get acquainted with a new definition of MASLD (metabolic dysfunction-associated steatotic liver disease) which replaces the previous NAFLD, but in my opinion readers urgently need an update of current available data according to the novel recommendations for such a complex context which could form the basis of new review article.

Minor comment:

Evidence suggests that IBD is a risk factor for the development of hepatic steatosis. Is there any evidence to state the opposite that NAFLD or hepatic steatosis is a risk factor for IBD? Any thoughts?

Comments on the Quality of English Language

The authors should carefully proof-read the article for typo and grammatical mistakes.

Author Response

This is an interesting and well written review article that focuses on providing an in-depth overview of the potential mechanisms that have been investigated but also highlight issues that need to be addressed for future studies in linking the pathophysiology of IBD and NAFLD. The authors have discussed in detail the prevalence of NAFLD in the general population and IBD patients, the clinical course of NAFLD in IBD patients and the mechanisms underlying the pathophysiology of NAFLD in IBD. However, the authors agreed that challenges still remain in understanding the link between NAFLD and IBD and despite a great number of studies investigating the association between these two disease conditions, data about the real burden of liver steatosis in IBD and the specific molecular mechanisms linking liver and gut inflammation are still confusing and less understood. The authors have further stated that due to the lack of standardized diagnostic tools, it becomes very difficult to rely on results from different studies that are based on evidence concerning prevalence and clinical course of the liver steatosis. The authors finally concluded that although various factors including pharmacological and surgical treatments, gut microbiota, parenteral nutrition, disease activity and duration could play a role on SLD (steatosis liver disease) occurrence in IBD patients, specific molecular pathways involved are not completely understood and many questions still remain unanswered. Moreover, readers will also get acquainted with a new definition of MASLD (metabolic dysfunction-associated steatotic liver disease) which replaces the previous NAFLD, but in my opinion readers urgently need an update of current available data according to the novel recommendations for such a complex context which could form the basis of new review article.

Minor comment: Evidence suggests that IBD is a risk factor for the development of hepatic steatosis. Is there any evidence to state the opposite that NAFLD or hepatic steatosis is a risk factor for IBD? Any thoughts?

R: We thank the reviewer for the comment. The role of NAFLD as risk factors for IBD represents an intriguing topic. However, to the best of our knowledge, there are currently no data proving a role of NAFLD or hepatic steatosis in the pathogenesis of IBD.